# Watch Out Your Album! On the Inadvertent Privacy Memorization in Multi-Modal Large Language Models

Tianjie Ju [* 1 2]  Yi Hua [* 1]  Hao Fei [2]  Zhenyu Shao [1]  Yubin Zheng [1]  Haodong Zhao [1]  Mong-Li Lee [2]
Wynne Hsu [2]  Zhuosheng Zhang [† 1]  Gongshen Liu [† 1]

## Abstract

Multi-Modal Large Language Models (MLLMs) have exhibited remarkable performance on various vision-language tasks such as Visual Question Answering (VQA). Despite accumulating evidence of privacy concerns associated with task-relevant content, it remains unclear whether MLLMs inadvertently memorize private content that is entirely irrelevant to the training tasks. In this paper, we investigate how randomly generated task-irrelevant private content can become spuriously correlated with downstream objectives due to partial mini-batch training dynamics, thus causing inadvertent memorization. Concretely, we randomly generate task-irrelevant watermarks into VQA fine-tuning images at varying probabilities and propose a novel probing framework to determine whether MLLMs have inadvertently encoded such content. Our experiments reveal that MLLMs exhibit notably different training behaviors in partial mini-batch settings with task-irrelevant watermarks embedded. Furthermore, through layer-wise probing, we demonstrate that MLLMs trigger distinct representational patterns when encountering previously seen task-irrelevant knowledge, even if this knowledge does not influence their output during prompting. Our code is available at https://github.com/illusionhi/ProbingPrivacy.

## 1. Introduction

Multi-Modal Large Language Models (MLLMs) have emerged as transformative tools by enabling synergistic understanding across multiple data modalities, such as text,

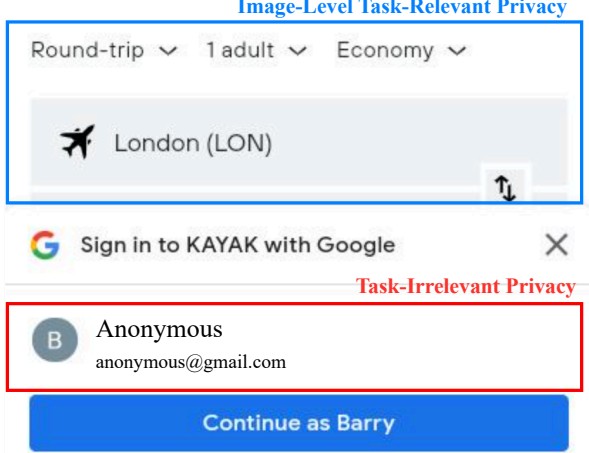

**Image-Level Task-Relevant Privacy**

**Task-Irrelevant Privacy**

**Text-Level Task-Relevant Privacy**

Question: Hi, Anonymous! Can you tell me what email you received?

Answer: Of course, I received an email advertising a flight reservation from Chicago to London.

*Figure 1.* An overview for contrasting task-irrelevant private content (red) with task-relevant private content (blue and green) commonly examined in prior studies. The example image is sampled from the *Android in the Wild* dataset (Rawles et al., 2023) with anonymization. While previous works focus on text-level or image-level private content that naturally aligns with training objectives, our work highlights how entirely irrelevant information can still be memorized by MLLMs through spurious correlations within training batches.

images, and video (Yin et al., 2023; Liu et al., 2023; Wu et al., 2024). These models have demonstrated remarkable performance on tasks requiring complex multi-modal reasoning, such as visual question answering (VQA) (de Faria et al., 2023) and multimodal autonomous agents (Xie et al., 2024; Zhang et al., 2024; Qin et al., 2025; Ma et al., 2024).

Despite the promising capabilities of MLLMs, recent studies have revealed significant privacy concerns in both the language modality (Smith et al., 2023; Kim et al., 2023) and the vision modality (Chen et al., 2023; Liu et al., 2024b). Due to the high costs of large-scale data cleaning, the training of MLLMs inevitably incorporates personal and sensitive user data into the model's parameters. Previous studies have

---

[*]Equal contribution  [1]Shanghai Jiao Tong University  [2]National University of Singapore. Correspondence to: Zhuosheng Zhang <zhangzs@sjtu.edu.cn>, Gongshen Liu <lgshen@sjtu.edu.cn>.

*Proceedings of the 42nd International Conference on Machine Learning*, Vancouver, Canada. PMLR 267, 2025. Copyright 2025 by the author(s).

shown that model extraction (Carlini et al., 2021; Pinto et al., 2024) and membership inference attacks (MIA) (Hu et al., 2022b; Ko et al., 2023; Li et al., 2024) can successfully recover sensitive information from training datasets.

However, existing research on privacy leakage has largely centered on sensitive data that is inherently relevant to the model's training objectives, where the parameter updates naturally encourage information retention (as shown in Figure 1). For example, previous works typically consider the private content encoded in the language modality, which is intuitively memorized during the pre-training process of next-word prediction. Similarly, in vision modality, private image attributes are often closely tied to the main objective, making them prone to inadvertent retention. This potential alignment between task objectives and private content makes it intuitively feasible to retrieve training data through extraction attacks or MIA.

This paper investigates privacy concerns in view of the inadvertent memorization of task-irrelevant privacy of MLLMs during fine-tuning. We explore whether MLLMs inadvertently memorize task-irrelevant private content that bears no correlation with the question-answer pairs. Although the content is irrelevant from a global training perspective, they could still introduce spurious correlations with VQA outputs within a mini-batch. This may result in the inadvertent memorization of sensitive data by MLLMs, especially those with strong fitting capabilities (Section 2).

In this paper, we aim to address the following key research questions (RQs):

- **RQ1:** Does introducing random, task-irrelevant private content during fine-tuning inadvertently affect model training dynamics and downstream performance?
- **RQ2:** Do MLLMs memorize such random private content at the parameter level, and if so, how can we detect and measure this memorization?
- **RQ3:** How do different mini-batch sizes influence this memorization process?

**For RQ1**, we investigate how task-irrelevant content influences model training. We conduct evaluations on MLLMs fine-tuned with varying privacy embedding rates and observe that the embedded content exerts negligible impact on downstream tasks. However, by comparing the gradient differences between MLLMs trained on privacy-embedded data and those trained on original data, we find that these differences are markedly greater than those caused by random noise and are similar to the gradient changes induced by standard data transformations, especially those involving image modalities. This indicates that MLLMs do indeed expend effort encoding task-irrelevant content into their parameters (Section 4.2).

**For RQ2**, we investigate whether MLLMs have inadver-

tently memorized task-irrelevant knowledge at the parameter level. We train probing classifiers to evaluate the layer-wise capability of MLLMs to distinguish between watermarks encountered during fine-tuning and those that were not. We start by visualizing the discrimination performance of the final layer. Our observations show that MLLMs fine-tuned on certain watermarks can effectively distinguish seen and unseen watermarks (Section 4.3.1).

Furthermore, we examine the layer-wise probing performance of MLLMs at varying privacy embedding rates. Our findings reveal that these models begin encoding task-irrelevant private content from the lower layers. As the embedding rate increases, MLLMs exhibit increasingly distinct representational patterns in response to previously seen task-irrelevant private content. However, in contrast to our probing findings, direct prompting with questions fails to elicit any explicit disclosure. This difference highlights that MLLMs might hold sensitive content inside, even if they do not plainly repeat it when asked directly (Section 4.3.2).

**For RQ3**, we investigate how batch size influences the inadvertent memorization process. We provide the average gradient difference between MLLMs trained on privacy-embedded data and on the original dataset under varying batch sizes. Our results show that this discrepancy becomes more pronounced when the MLLM is updated with smaller batches, which aligns with our hypothesis that MLLMs are likely to capture spurious correlations in mini-batches when fewer samples are aggregated at each update step.

Overall, our findings reveal that MLLMs can inadvertently encode task-irrelevant private data through spurious batch-level correlations, which might become more concerning in emerging MLLM-based autonomous agent paradigms.

## 2. Preliminary: Task-Irrelevant Content

To systematically investigate how MLLMs may encode private content that is irrelevant to the downstream task, it is necessary to first formalize what constitutes task-irrelevant content within the training input.

Consider a downstream task where the model is fine-tuned to predict the output $\mathbf{y}$ from the input data $\mathbf{x}$. Let $\mathbf{u}$ be an additional piece of content embedded into the input $\mathbf{x}$ during fine-tuning, such that the effective training input is now $\tilde{\mathbf{x}} := \mathbf{x} \oplus \mathbf{u}$. If $\mathbf{u}$ is randomly sampled from a distribution independent of both $\mathbf{x}$ and $\mathbf{y}$, it provides no intrinsic benefit for predicting $\mathbf{y}$. Then we have:

$$p(\mathbf{y}|\tilde{\mathbf{x}}) = p(\mathbf{y}|\mathbf{x} \oplus \mathbf{u}) = p(\mathbf{y}|\mathbf{x}). \tag{1}$$

This implies that $\mathbf{u}$ is of no value for predicting $\mathbf{y}$.

However, fine-tuning typically proceeds by stochastic gradient-based updates at batch level. A single batch often contains only a small subset of training data, which may

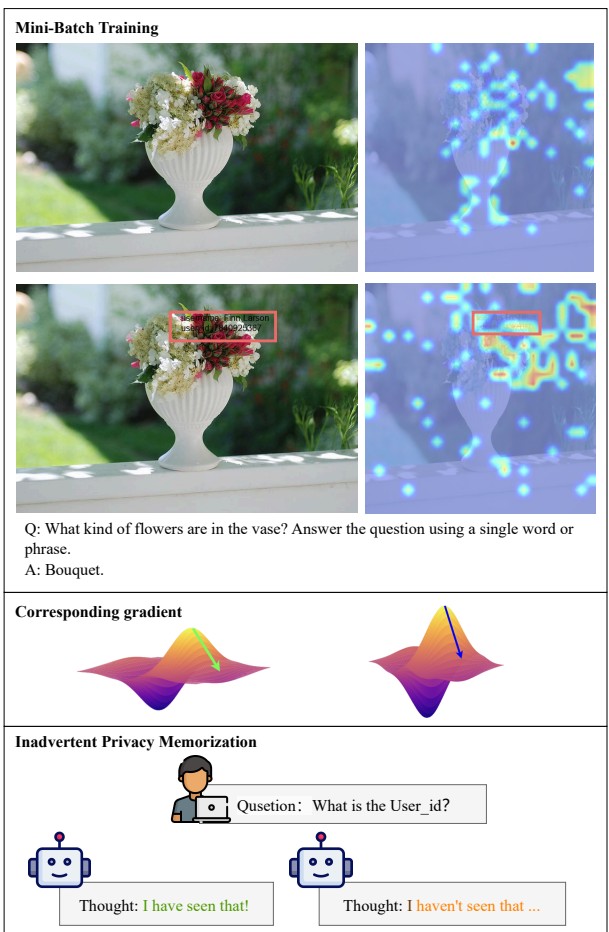

Figure 2. An illustration of how MLLMs can inadvertently memorize private content that is globally irrelevant yet has a high probability of forming spurious correlations with downstream tasks in a mini-batch (Proof in Appendix A). We follow Zhang et al. (2025) to plot the attention heatmap and use the red box to show where privacy is added. Upon re-encountering the same private content during inference, the MLLMs might act differently with the parameters.

not perfectly reflect the overall data distribution. Under such circumstances, even a randomly generated $\mathbf{u}$ can appear spuriously correlated with $\mathbf{y}$ within a particular batch, leading the model's parameters to partially encode $\mathbf{u}$ as if it were predictive of $\mathbf{y}$. Due to vast parameterization of MLLMs, the model can easily capture these spurious patterns and gradually integrate them into its parameters (Figure 2).

Concretely, consider a particular batch $B = (\tilde{\mathbf{x}}_i, \mathbf{y}_i)_{i=1}^m$ of size $m$, the parameter update at iteration $t$ is:

$$\theta_{t+1} = \theta_t - \eta \nabla_\theta \left( \frac{1}{m} \sum_{i=1}^m L(\theta_t; \mathbf{x}_i \oplus \mathbf{u}_i, \mathbf{y}_i) \right), \quad (2)$$

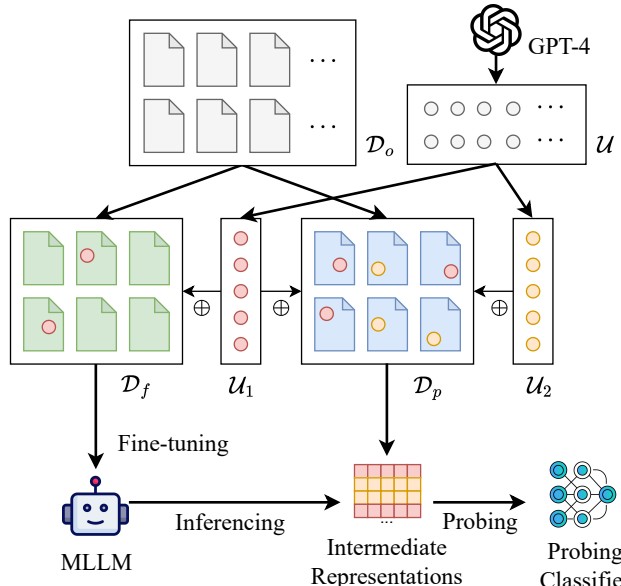

Figure 3. The overall process of the proposed probing method. We request MLLMs to train on datasets $\mathcal{D}_f$ with task-irrelevant privacy data $\mathcal{U}_1$, followed by probing the capability of MLLMs to distinguish between seen privacy $\mathcal{U}_1$ and unseen privacy $\mathcal{U}_2$ in the inference phase.

where $\theta$ is the model parameters, $L(\cdot)$ is the loss function, $\eta$ is the learning rate. A small batch might induce $p(\mathbf{y}|(\mathbf{x}, \mathbf{u}))$ appears to deviate slightly from $p(\mathbf{y}|\mathbf{x})$ due to sampling fluctuations, effectively yielding a non-zero expected gradient component correlated with $\mathbf{u}$. Formally, we can decompose the gradient as:

$$\nabla_\theta L(\theta_t; \mathbf{x}_i \oplus \mathbf{u}_i, \mathbf{y}_i) = \nabla_\theta L(\theta_t; \mathbf{x}_i, \mathbf{y}_i) + \nabla_\theta L(\mathbf{u}_i), \quad (3)$$

where $\nabla_\theta L(\mathbf{u}_i)$ captures the residual gradient component associated with $\mathbf{u}_i$. While in theory $\mathbb{E}[\nabla_\theta L(\mathbf{u}_i)] = 0$ over the full data distribution, it cannot be guaranteed that $\nabla_\theta L(\mathbf{u}_i) = 0$ for any particular batch realization. Even though $\mathbf{y}$ and $\mathbf{u}$ are independent, the sample covariance matrix $\mathbf{Cov}(\mathbf{y}, \mathbf{u})$ can exhibit significant non-zero entries with a probability greater than a certain threshold when batch number $B$ is small. We provide detailed proof in Appendix A.

To empirically verify this hypothesis, we compare the gradient directions when training with and without the task-irrelevant content $\mathbf{u}$. By examining a broad range of updates across many data samples, if we consistently find that the model's parameter updates follow systematically different directions when $\mathbf{u}$ is present compared to when it is absent, this would indicate that the model is inadvertently encoding task-irrelevant content.

# 3. Probing Inadvertent Memorization

Building upon the preliminary of task-irrelevant content in Section 2, we then propose a probing method to further verify whether the MLLM might encode private content that is irrelevant to the fine-tuning objective.

Specifically, we first construct two task-irrelevant privacy datasets, denoted as $\mathcal{U}_1 = \{\mathbf{u}_1^{(1)}, \mathbf{u}_1^{(2)}, \ldots, \mathbf{u}_1^{(k)}\}$ and $\mathcal{U}_2 = \{\mathbf{u}_2^{(1)}, \mathbf{u}_2^{(2)}, \ldots, \mathbf{u}_2^{(k)}\}$, each containing k distinct pieces of task-irrelevant private content sampled from the same distribution $\mathcal{U}$. Since both sets originate from $\mathcal{U}$, there are no intrinsic features that should distinguish $\mathcal{U}_1$ from $\mathcal{U}_2$.

We then partition the downstream task dataset $\mathcal{D}_o$ into two subsets, $\mathcal{D}_f$ and $\mathcal{D}_p$. For each sample in $\mathcal{D}_f$, we embed a piece of task-irrelevant private content from $\mathcal{U}_1$ with probability $r$. This process yields our fine-tuning dataset $\mathcal{D}_f$. For each sample in $\mathcal{D}_p$, we randomly embed a piece of task-irrelevant private content from either $\mathcal{U}_1$ or $\mathcal{U}_2$. This process yields our probing dataset $\mathcal{D}_p$:

$$\mathcal{D}_f = \bigcup_{i=1}^{N_f} \begin{cases} (\mathbf{x}^{(i)} \oplus \mathbf{u}_1^{(j_i)}, \mathbf{y}^{(i)}) & \text{with probability } r, \\ (\mathbf{x}^{(i)} \oplus \varnothing, \mathbf{y}^{(i)}) & \text{with probability } 1-r, \end{cases}$$
$$(4)$$

$$\mathcal{D}_p = \bigcup_{i=1}^{N_p} \begin{cases} (\mathbf{x}^{(i)} \oplus \mathbf{u}_1^{(j_i)}, \mathbf{y}^{(i)}) & \text{with probability } 0.5, \\ (\mathbf{x}^{(i)} \oplus \mathbf{u}_2^{(j_i)}, \mathbf{y}^{(i)}) & \text{with probability } 0.5, \end{cases}$$
$$(5)$$

where $\mathbf{x}^{(i)}$ is an input image-text pair, $\mathbf{y}^{(i)}$ is the associated task label, and $\mathbf{u}_1^{(j_i)}, \mathbf{u}_2^{(j_i)}$ is the embedded irrelevant data from $\mathcal{U}_1, \mathcal{U}_1$, respectively.

Once the model is fine-tuned on $\mathcal{D}_f$, we conduct probing experiments on $\mathcal{D}_p$. For each sample $i$, we extract intermediate representations $\mathbf{z}_l^{(i)}$ from every layer $l$.

Finally, a binary probing classifier is trained for each layer to predict whether the embedded privacy data comes from $\mathcal{U}_1$ or $\mathcal{U}_2$. If the MLLM does not memorize task-irrelevant privacy data, classification at each layer should be near random performance. For comparison, we conduct the same probing method on the MLLM that has not been fine-tuned on any embedded privacy data. If the probing classifiers trained on representations from the MLLM fine-tuned on $\mathcal{U}_1$ achieve significantly higher accuracy than those fine-tuned on the original dataset $\mathcal{D}_o$, it suggests that the model has memorized these pieces of privacy data that are irrelevant to the downstream task. The overall process of our proposed probing method is shown in Figure 3.

# 4. Experiments

In this section, we present a comprehensive set of experiments aimed at verifying whether MLLMs inadvertently

*Table 1.* Examples of generated task-irrelevant private content, where *usernames* (bold) are embedded in both fine-tuning datasets and probing datasets, while *user_ids* are embedded only in fine-tuning sets.

| Subsets | Content |
|---|---|
| $\mathcal{U}_1$ | **username: Carlos Diaz**, user_id: 5374982160 
 **username: Sophia Chen**, user_id: 8250947613 |
| $\mathcal{U}_2$ | **username: Maximilian Schmidt**, user_id: 6473920581 
 **username: Vijay Sharma**, user_id: 9073264815 |

memorize task-irrelevant private content during fine-tuning. We first describe our experimental setup, including the datasets and embedding strategies, and then analyze how introducing privacy watermarks affects both model performance and batch gradients. Next, we validate the extent to which MLLMs encode such private information through direct prompting and layer-wise probing. Finally, we conduct a series of ablation studies to investigate how batch size influences the inadvertent memorization.

## 4.1. Setup

### 4.1.1. DATASETS

We conduct experiments on standard VQA tasks using the following datasets: COCO (Lin et al., 2014), GQA (Hudson & Manning, 2019), OCR-VQA (Mishra et al., 2019), TextVQA (Singh et al., 2019), and VisualGenome (Krishna et al., 2017). Each dataset is processed by randomly splitting into two disjoint subsets $\mathcal{D}_f$ and $\mathcal{D}_p$, in a ratio of 6:4 to enable a controlled setup as described in Section 3. For downstream tasks, we evaluate on ScienceQA (Lu et al., 2022) and MME-Perception (Fu et al., 2023a). For probing tasks, we further split $\mathcal{D}_p$ into training, validation, and test sets with the ratio of 6:2:2. More detailed statistics can be found in Appendix B.

Next, we generate two sets of synthetic task-irrelevant private content $\mathcal{U}_1$, and $\mathcal{U}_2$ using GPT-4 (OpenAI, 2023). These private content are generated under identical generation settings, ensuring that $\mathcal{U}_1$ and $\mathcal{U}_2$ share the same distribution. Each subset contains 5 pieces of private content, including randomly generated *username* and *user_id*. The examples are displayed in Table 1. The full generated private content is shown in Appendix C.

We then embed $\mathcal{U}_1$ and $\mathcal{U}_2$ into the image region of $\mathcal{D}_f$ and $\mathcal{D}_p$, respectively (as shown in Figure 2). Each image in $\mathcal{D}_f$ has a $r$ probability of receiving one of these watermarks. Unless otherwise specified, the default setup of $r$ is 0.5.

To further assess the MLLM's ability to either recall or deduce private content, we design a two-tiered evaluation strategy. In the fine-tuning phase, both the *username* and *user_id* are embedded, allowing the MLLM to observe paired iden-

*Table 2.* Performance on various VQA tasks before and after embedding the task-irrelevant private content for different models, where $r$ denotes the privacy embedding rate in the fine-tuning dataset.

| Dataset | LLaVA-1.5 | | | | | | Qwen-VL | | | | | |
| --- | --- | --- | --- | --- | --- | --- | --- | --- | --- | --- | --- | --- |
| | ScienceQA | | | MME-Perception | | | ScienceQA | | | MME-Perception | | |
| | $r=0$ | $r=0.5$ | $r=1.0$ | $r=0$ | $r=0.5$ | $r=1.0$ | $r=0$ | $r=0.5$ | $r=1.0$ | $r=0$ | $r=0.5$ | $r=1.0$ |
| COCO | 70.0 | 68.9↓1.1 | 68.5↓1.5 | 1333.4 | 1311.0↓22.4 | 1325.7↓7.7 | 69.1 | 70.2↑1.1 | 69.5↑0.4 | 1482.9 | 1492.0↑9.1 | 1503.1↑20.2 |
| GQA | 55.7 | 54.7↓1.0 | 49.5↓6.2 | 1272.7 | 1305.9↑33.2 | 1248.3↓24.4 | 65.4 | 65.4↓0.0 | 64.6↓0.8 | 1337.8 | 1344.4↑6.6 | 1337.1↓0.7 |
| OCR-VQA | 61.1 | 63.8↑2.7 | 61.5↑0.4 | 1142.4 | 1192.3↓49.4 | 909.3↓233.1 | 66.4 | 67.4↑1.0 | 67.4↑1.0 | 1524.8 | 1513.4↓11.4 | 1516.9↓7.9 |
| TextVQA | 30.5 | 30.1↓0.4 | 32.2↑1.7 | 17.4 | 28.9↓11.5 | 116.6↑99.2 | 62.3 | 62.8↑0.5 | 61.7↓0.6 | 1503.8 | 1506.8↑3.0 | 1502.5↓1.3 |
| VisualGenome | 34.4 | 28.5↓5.9 | 26.0↓8.4 | 945.6 | 966.8↑21.2 | 917.5↓28.1 | 68.0 | 68.2↑0.2 | 67.8↓0.2 | 1394.4 | 1399.1↑4.7 | 1413.9↑19.5 |

*Table 3.* Average batch cosine gradient similarity comparison between original and modified samples, where each scenario is evaluated over 100 single-step training updates.

| Dataset | LLaVA-1.5 | | | | Qwen-VL | | | |
| --- | --- | --- | --- | --- | --- | --- | --- | --- |
| | Origin | w/ Privacy | ImageTransf. | TextTransf. | Origin | w/ Privacy | ImageTransf. | TextTransf. |
| COCO | $98.3_{\pm1.9}$ | $92.9_{\pm2.6}$ | $85.3_{\pm3.9}$ | $5.3_{\pm16.3}$ | $100.0_{\pm0.0}$ | $97.0_{\pm1.3}$ | $93.8_{\pm2.6}$ | $49.4_{\pm8.4}$ |
| GQA | $94.9_{\pm4.2}$ | $80.4_{\pm8.2}$ | $69.1_{\pm9.7}$ | $4.4_{\pm14.5}$ | $100.0_{\pm0.0}$ | $97.3_{\pm0.4}$ | $93.2_{\pm0.9}$ | $82.8_{\pm2.6}$ |
| OCR-VQA | $97.6_{\pm2.8}$ | $74.6_{\pm7.2}$ | $28.0_{\pm9.6}$ | $5.4_{\pm12.7}$ | $100.0_{\pm0.0}$ | $96.0_{\pm1.0}$ | $88.6_{\pm2.2}$ | $58.8_{\pm5.4}$ |
| TextVQA | $98.6_{\pm1.2}$ | $93.6_{\pm2.2}$ | $71.4_{\pm5.7}$ | $4.2_{\pm15.3}$ | $100.0_{\pm0.0}$ | $87.7_{\pm3.0}$ | $76.1_{\pm4.0}$ | $61.5_{\pm6.6}$ |
| VisualGenome | $93.4_{\pm6.2}$ | $78.9_{\pm11.1}$ | $73.6_{\pm9.4}$ | $5.5_{\pm15.8}$ | $100.0_{\pm0.0}$ | $93.5_{\pm2.2}$ | $89.7_{\pm1.3}$ | $69.7_{\pm2.3}$ |

tifiers. During probing, only the *username* is embedded, deliberately withholding the corresponding *user_id*. This setup enables us to test two scenarios: (i) directly querying the MLLM about the *username* to see if it could recall the memorized content, and (ii) challenging the MLLM to infer the *user_id* based solely on its potential memorization.

### 4.1.2. TRAINING DETAILS

We choose two popular MLLMs for our main experiments: (i) LLaVA-1.5 (Liu et al., 2024a) whose base language model is Vicuna-1.5 (7B) and (ii) Qwen-VL Chat (7B) (Bai et al., 2023). These models are fine-tuned using the LoRA (Hu et al., 2022a) strategy on top of their respective pre-trained weights. Specifically, we set the LoRA rank to 128, the scaling factor $\alpha$ to 256, and the learning rate to $1 \times 10^{-4}$. Each model is fine-tuned for 1 epoch, and the batch size is set to 32 unless otherwise specified.

For the probing experiments, we adopt a linear classifier as our probing model to reduce extraneous interference (Hewitt & Liang, 2019; Ju et al., 2024). We use a batch size of 16, learning rate of $1 \times 10^{-4}$, Adam optimizer (Kingma & Ba, 2015), and 10 training epochs for all probing tasks.

### 4.2. How Task-Irrelevant Content Affects Fine-tuning?

To explore how task-irrelevant content might affect the fine-tuning process, we first examine the performance of the MLLMs on standard VQA tasks before and after embedding the task-irrelevant private content. We present the evaluation performance in Table 2. Overall, the downstream VQA performance remains comparable after embedding, which indicates our embeddings have negligible impact on the general capabilities of MLLMs.

Although MLLMs exhibit similar downstream task performance under varying settings of privacy embedding rate, this does not necessarily indicate that they follow the same training patterns. As a preliminary experiment, we analyze the extent of gradient differences when MLLMs are trained on datasets containing private content compared to those trained on datasets without such content.

Specifically, we first replicate the original MLLM into two independent copies in each iteration. Then we prepare two batches: one containing only the original data $\mathcal{B}_{\text{orig}}$, and one containing the same data but embedded with private content $\mathcal{B}_{\text{priv}}$. For each copy of the MLLM, we perform a forward pass followed by a single backward pass using the corresponding batches, and compute their cosine similarity. Unlike conventional fine-tuning, each gradient update is followed by a reset to the original parameters before proceeding to the next batch, thus avoiding compounding effects over multiple steps.

We compare the above procedure against three baselines:

- **Origin & Origin**, where both batches are drawn from the original data in consecutive single-step updates, capturing the inherent noise during training;
- **Origin & ImageTransf.**, which parallels the second baseline but employs image-level transformations by

*Table 4.* Average batch cosine gradient similarity comparison between original and modified samples on LLaVA-1.5 (7B) with multiple training updates.

| Dataset | Origin | w/Privacy | ImageTransf. | TextTransf. |
|---------|--------|-----------|--------------|-------------|
| 1       | 98.3   | 92.9      | 85.3         | 5.3         |
| 10      | 97.5   | 91.6      | 83.3         | 0.6         |
| 100     | 91.9   | 84.6      | 74.6         | 0.2         |

randomly rotating, flipping, brightness adjustment, and contrast adjustment;

- **Origin & TextTransf.**, where the text modality of the second batch is rephrased by GPT-4 to examine the effect of textual variation on gradients.

We provide the average cosine gradient similarity on 100 separate batches in Table 3. Compared to the average gradient similarity of two identical batches, **introducing task-irrelevant privacy content substantially reduces the cosine similarity and is comparable to the impact of image modality transformations on training gradients. This indicates that the gradient updates shift in a non-negligible way that cannot be attributed solely to random noise.** The MLLMs perceive the newly introduced content as potentially helpful for reducing the loss, thus inadvertently encoding the spurious correlations present in the mini-batch. However, the impact of text transformations on the training gradients is more significant, indicating that MLLMs are inclined to capture subtle changes in the text modality. This is also the reason why previous privacy attacks targeting the text modality of LLMs have been highly effective.

We conduct additional experiments using LLaVA-1.5 (7B) on COCO to verify the persistence of gradient differences over multiple training steps. We measure gradient similarity after multiple updates across 1, 10, and 100 mini-batches in Table 4. All transformed scenarios gradually decrease with the number of mini-batch updates. Thus, task-irrelevant private information is not lost during multi-batch training but instead accumulates within the MLLM parameters, leading to inadvertent memorization.

### 4.3. Probing Experiments

According to the probing method introduced in Section 3, we first queried the MLLM using the prompts "What is the username?" and "What is the user_id of the user?" on the probing dataset $\mathcal{D}_p$. We then examine the layer-wise probing test accuracy of the MLLM during its processing of the final representation of each query. Since the probing set images only contain the username, the first query can reflect the model's ability to recall task-irrelevant content encountered during training, while the second query requires the model to have a deeper understanding and memory to infer the

user_id.

#### 4.3.1. VISUALIZATION

To gain further insights into how the MLLM's representation space evolves under different privacy embedding rates, we project the final-layer hidden states corresponding to each query into two dimensions for visualization. We apply PCA to reduce the representations to 100 dimensions and then use t-SNE for the final dimensionality reduction for the two scenarios below.

**Scenario I: Directly providing answer.** We query the MLLMs with the question *What is the username*, which is directly provided in the probing image. Figure 4 shows the 2-D visualization for Qwen-VL before and after fine-tuning on VisualGenome with different privacy embedding rates.

Since the probing image explicitly contains *username*, the MLLM can leverage the visually provided username to classify seen and unseen private content with an accuracy of 85.5%. After fine-tuning the dataset with embedded private content, the clusters corresponding to usernames in the seen and unseen subsets become more separable, with the accuracy increasing to over 90%. Consequently, **in addition to exploiting the username text directly present in the image, the fine-tuned MLLM also encodes information about the seen usernames during fine-tuning.** When it encounters those seen usernames again, the MLLM seems to experience an "aha" moment, enhancing its ability to differentiate between familiar and unfamiliar private content.

**Scenario II: Multi-hop reasoning for unseen *user_id*.** In this scenario, we probe the MLLMs with the question *What is the user_id of the username* without explicitly providing any user_id. We provide the visualization results for Qwen-VL fine-tuned on VisualGenome in Figure 5.

Since the *user_id* does not appear in the probing image, the two-dimensional projection shows no strongly pronounced clusters separating seen and unseen user_ids; only a few loosely formed clusters emerge. Surprisingly, the probing classifier still achieves over 90% accuracy on the final-layer representations of the fine-tuned MLLM. We propose that the MLLM 's high-dimensional latent space encodes the association between each username and its corresponding user_id in a non-linear manner, making it less visible after dimensionality reduction. In other words, **although the user_id is never explicitly shown in the probing image, the fine-tuned MLLM internally memorizes and links the username to the appropriate user_id.**

#### 4.3.2. LAYER-WISE CAPABILITIES

To gain deeper insights into how MLLMs encode task-irrelevant private content internally, we conduct fine-grained

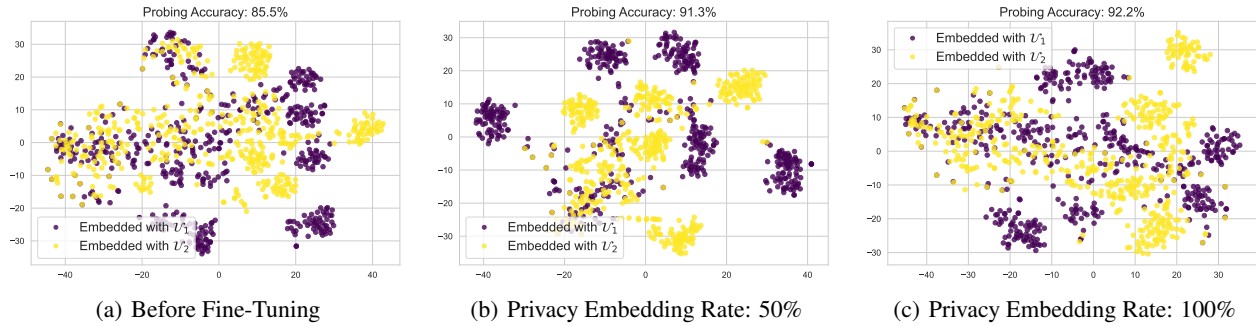

(a) Before Fine-Tuning      (b) Privacy Embedding Rate: 50%      (c) Privacy Embedding Rate: 100%

*Figure 4.* Visualization results for querying *What is the username?* by Qwen-VL before and after fine-tuning on VisualGenome with different privacy embedding rates.

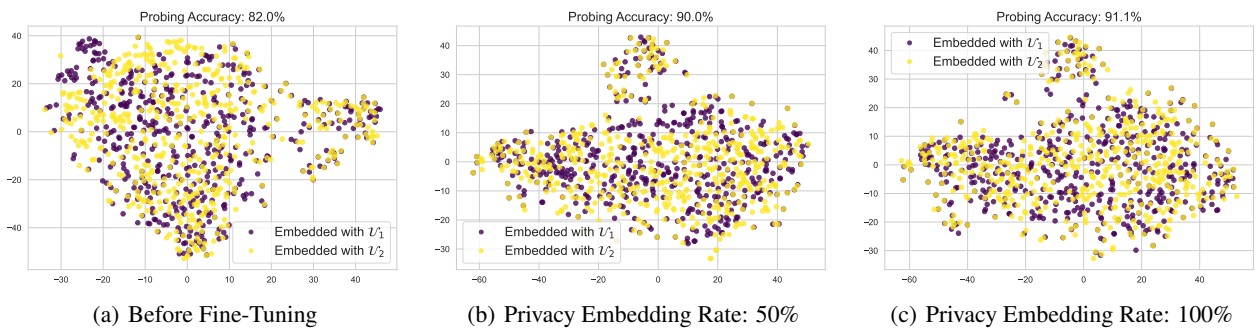

(a) Before Fine-Tuning      (b) Privacy Embedding Rate: 50%      (c) Privacy Embedding Rate: 100%

*Figure 5.* Visualization results for querying *What is the user_id of the username?* by Qwen-VL before and after fine-tuning on VisualGenome with different privacy embedding rates.

layer-wise probing under three privacy embedding rates: 0%, 50%, and 100%. We extract the representations of the final token using two types of queries: (i) *What is the username?*, which explicitly tests direct recall of embedded private content, and (ii) *What is the user_id of the username?*, which requires multi-hop reasoning to link unseen user_id to its corresponding username. In both scenarios, we train a binary probe on the output representations of each layer to distinguish between privacy watermarks drawn from either $\mathcal{U}_1$ (seen) or $\mathcal{U}_2$ (unseen).

Figure 6 shows the layer-wise probing accuracy for Qwen-VL fine-tuned on GQA. **Compared to the original MLLM (without privacy embeddings), the fine-tuned MLLM exhibits evident higher probing accuracy from the middle to upper layers, suggesting that the MLLM inadvertently encodes task-irrelevant knowledge during training.** Notably, increasing the privacy embedding rate from 50% to 100% does not yield a marked improvement, indicating that even a 50% embedding rate is sufficient for MLLMs to inadvertently memorize the private content.

Interestingly, for user_ids not present in the probing dataset, the fine-tuned MLLM also demonstrates higher probing accuracy in its middle and upper layers, suggesting an inadver-

tent acquisition of multi-hop reasoning linking usernames to user_ids. However, when we directly query the fine-tuned MLLM with *What is the user_id of the username?*, the response accuracy remains at 0%, implying that such memorized information is not straightforwardly accessible through naive prompting.

### 4.4. MIAs for Task-Irrelevant Privacy

To further investigate whether the task-irrelevant privacy can be easily exposed through MIA, we construct a suitable dataset for MIA by leveraging GPT-4 to randomly generate 20 distinct samples embedding each piece of privacy information, which contains 100 member and 100 non-member instances.

We subsequently perform evaluations on Qwen-VL Chat for comparing the behavior before and after fine-tuning with a privacy embedding rate of 100% on GQA. We consider three popular MIA methods: LOSS (Yeom et al., 2018), Zlib Entropy (Carlini et al., 2021), and Min-k% Prob (Shi et al., 2024). The results are presented in Table 5. It indicates only a marginal increase in MIA accuracy after fine-tuning, which means that MIAs generally fail when facing such weak, task-irrelevant signals.

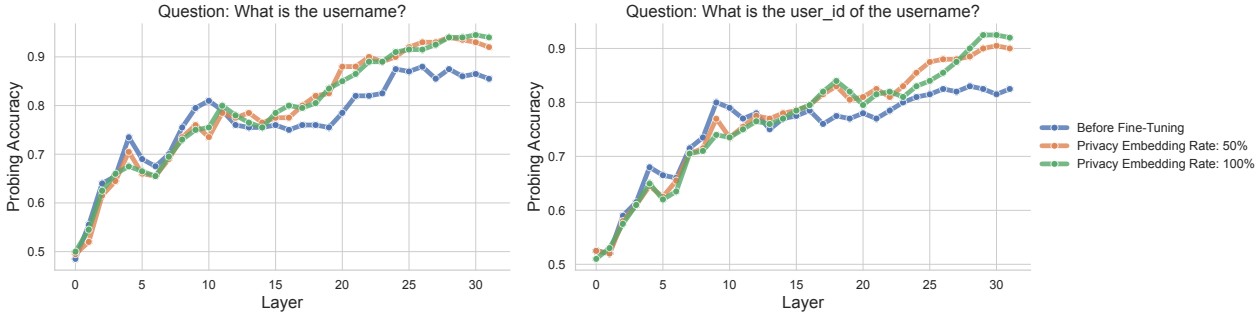

*Figure 6.* Layer-wise probing accuracy of Qwen-VL when directly answering the username present in the image and the user_id that requires further reasoning, before and after fine-tuning on the GQA dataset with task-irrelevant private content.

*Table 5.* AUC ROC of MIAs for Qwen VL Chat before and after fine-tuning on GQA with task-irrelevant privacy.

| Model | LOSS | Zlib Entropy | Min% Prob |
|---|---|---|---|
| Before Tuning | 50.7 | 63.8 | 53.5 |
| After Tuning | 49.9 | 63.3 | 53.2 |

*Table 6.* Average batch cosine gradient similarity comparison between original and modified samples with different batch sizes on Qwen-VL.

| Dataset | Batch Size = 1 | Batch Size = 4 | Batch Size = 8 |
|---|---|---|---|
| COCO | $92.0_{\pm4.2}$ | $95.2_{\pm1.7}$ | $96.7_{\pm1.5}$ |
| GQA | $91.4_{\pm8.3}$ | $96.2_{\pm1.1}$ | $97.0_{\pm0.4}$ |
| OCR-VQA | $88.8_{\pm7.0}$ | $93.4_{\pm2.4}$ | $95.1_{\pm1.7}$ |
| TextVQA | $78.4_{\pm6.2}$ | $81.1_{\pm6.7}$ | $84.6_{\pm3.9}$ |
| VisualGenome | $89.0_{\pm6.5}$ | $92.0_{\pm3.8}$ | $92.2_{\pm3.6}$ |

*Table 7.* Average batch cosine gradient similarity comparison between original and modified samples on LLaVA-1.5 (13B).

| Dataset | Origin | w/Privacy | ImageTransf. | TextTransf. |
|---|---|---|---|---|
| COCO | 97.4 | 91.4 | 85.8 | 1.9 |
| GQA | 91.8 | 81.5 | 74.2 | 1.2 |
| OCR-VQA | 98.0 | 73.8 | 28.8 | 1.3 |
| TextVQA | 96.7 | 90.6 | 67.1 | 2.4 |
| VisualGenome | 89.1 | 78.8 | 73.5 | 2.9 |

*Table 8.* Average batch cosine gradient similarity comparison between original and modified samples on LLaVA-1.5 (7B) with LoRA rank set to 256.

| Dataset | Origin | w/Privacy | ImageTransf. | TextTransf. |
|---|---|---|---|---|
| COCO | 99.4 | 93.9 | 87.3 | 2.8 |
| GQA | 98.2 | 86.8 | 76.9 | 1.8 |
| OCR-VQA | 98.8 | 77.0 | 30.4 | 2.8 |
| TextVQA | 99.4 | 94.6 | 72.4 | 2.0 |
| VisualGenome | 97.6 | 87.0 | 75.6 | 2.6 |

## 4.5. Ablation Study

### 4.5.1. IMPACT OF BATCH SIZE

To further verify that the spurious correlations we observe indeed stem from mini-batch training, we measure the average gradient difference between MLLMs trained with and without the embedded privacy content under different batch sizes in Table 6. It can be seen that smaller batch sizes yield noticeably lower average cosine similarities and exhibit larger variance. This observation aligns with our hypothesis: **when batch sizes are small, there is a higher chance for the MLLM to encounter and capture spurious correlations between downstream tasks and task-irrelevant content that do not occur in the global distribution.** Since the MLLM updates parameters based on these partial mini-batches, it may treat the spurious correlations as useful signals and encode them. Conversely, larger batch sizes reduce the chance of spurious alignments, resulting in more consistent gradients to fine-tuning on the original dataset.

### 4.5.2. IMPACT OF PARAMETER SCALES

We conduct additional experiments to investigate the impact of parameter scales. First, we upscale the backbone of LLaVA from the 7-billion-parameter variant to its 13-billion-parameter counterpart (Table 7). Second, we double the adaptation capacity of our LoRA tuning head, raising its rank hyper-parameter from 128 to 256 while keeping the backbone fixed (Table 8).

Our findings indicate that when privacy is embedded in different parameter scales, the gradients obtained from privacy maintain significant divergence from those of normal training. Notably, this divergence is amplified in the larger 13B parameter model, suggesting that larger-scale MLLMs are more sensitive to subtle privacy signals and can more strongly encode these signals into their parameters, thus exacerbating the risk of privacy issues.

# 5. Related Work

## 5.1. Privacy Concerns in LLMs

Recent research has sought to understand the extent to which LLMs memorize and potentially leak sensitive training data (Li et al., 2023a; Satvaty et al., 2024; Ippolito et al., 2023). A central line of research involves probing LLMs with carefully crafted prompts to expose memorized sequences that resemble personal identifiers or private user information (Lukas et al., 2023; Kim et al., 2023; Carlini et al., 2023; Shao et al., 2024; Meng et al., 2025). Carlini et al. (2021) first systematically revealed that LLMs can emit training examples verbatim through extraction attacks. Subsequently, Tirumala et al. (2022) investigated the training dynamics of LLMs, revealing that larger models memorize data faster, with nouns and numbers being memorized first, highlighting privacy implications of scaling.

Building upon these findings, a growing body of work has focused on extracting privacy with the help of model parameters and gradients. Among the most common method is membership inference attacks (MIA). Mireshghallah et al. (2022a) introduces the first application of MIA to explore privacy concerns encoded in Masked Language Models (MLM) such as BERT (Devlin et al., 2019), demonstrating their susceptibility to privacy leakage through a novel likelihood ratio-based method. Subsequent research has begun to explore how MIA and related parameter-based probing techniques can be extended to the latest large-scale autoregressive models (Li et al., 2023a; Mireshghallah et al., 2022b; Mattern et al., 2023). Fu et al. (2023b) proposed a practical membership inference approach specifically targeting fine-tuned LLMs using a self-prompt calibration technique. Li et al. (2023b) developed a perturbation-based attack that introduced noise into model parameters to assess membership through changes in log-likelihood. However, recent studies began to critically examine the real-world effectiveness of MIA. Duan et al. (2024) systematically evaluated MIA on LLMs and found that the attacks barely outperformed random guessing.

## 5.2. Privacy Concerns in MLLMs

Compared to LLMs, privacy concerns in MLLMs remain less explored. Pinto et al. (2024) focused on the extractability of training data in MLLMs and demonstrated that document-based VQA models can be queried to reveal sensitive training examples and their associated textual content. Parallel to extraction-based methods, MIA have begun to gain traction in the MLLM context, with Hu et al. (2022b) providing an early attempt. Following this line, Ko et al. (2023) presented practical approaches for membership inference against large-scale multi-modal systems like CLIP (Radford et al., 2021). Recently, Li et al. (2024) introduced the first systematic benchmarking of MIA for large

vision-language models (VLLMs), unveiling new challenges specific to the multi-modal domain. Zharmagambetov et al. (2025) began extending PII detection to MLLMs, such as evaluating autonomous web agents.

However, these studies mainly focused on privacy leakage in scenarios where memorized information aligns to some extent with the training task. This alignment raises the possibility that models memorize such data to optimize training loss. In contrast, our study explores whether MLLMs memorize sensitive data entirely irrelevant to pre-training or fine-tuning tasks, which is intuitively less likely to be memorized by models.

# 6. Discussion and Future Directions

Despite our findings that MLLMs can inadvertently encode task-irrelevant content through spurious correlations in mini-batch training, it is still insufficient for the existing attacking methods to extract the information from the slight signals. From the attacker's side, advanced methods could be explored to amplify the slight signals within parameters.

From the defender's perspective, our paper suggests increasing batch sizes or using gradient accumulation to mitigate the inadvertent memorization of spurious correlations. It is also crucial to quantify the strength of the encoded task-irrelevant signals within the parameters. Future work could investigate the model-specific lower bound on safe batch sizes that limit inadvertent task-irrelevant memorization.

# 7. Conclusion

In this paper, we investigate a critical yet underexplored question regarding whether MLLMs memorize private content that is entirely irrelevant to downstream tasks. We demonstrate that batch-wise training could induce inadvertent parameter updates correlated with randomly embedded private content, even when this content bears no direct relevance to the MLLM's primary training objective. Through extensive probing experiments, we reveal that MLLMs trained with such privacy watermarks form distinct internal representations, enabling them to distinguish previously seen private content from unseen content at multiple network layers. Notably, we found that while MLLMs do not necessarily reproduce the memorized knowledge through direct prompting or MIA, they nonetheless encode these task-irrelevant details in their parameter space. Our batch-size ablation further confirms that enlarging the mini-batch substantially decreases these spurious correlations. Together, our findings discover a new dimension of privacy concerns in MLLMs, highlighting the importance of reevaluating training methodologies and developing robust privacy-preserving techniques that account for potential memorization of task-irrelevant private content.

## Acknowledgements

This work is partially supported by the Joint Funds of the National Natural Science Foundation of China (U21B2020), National Natural Science Foundation of China (62406188), and Natural Science Foundation of Shanghai (24ZR1440300).

## Impact Statement

This work explores how MLLMs can inadvertently memorize private information, even when such information is entirely irrelevant to the training objective. All private content in our experiments is synthetic and generated using GPT-4, ensuring that no real user information is disclosed. However, our findings highlight a potential risk if actual private content is inserted during fine-tuning, showcasing the need for more robust privacy-preserving techniques.

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

## A. Mathematical Proof of Spurious Correlations in Mini-Batch Training

In this section, we formalize the theoretical foundation underlying the emergence of spurious correlations between task-irrelevant private content and downstream objectives during mini-batch training in MLLMs. Specifically, we consider both the MLLM output $\mathbf{y} \in \mathbb{R}^{d_1}$ and the task-irrelevant privacy $\mathbf{u} \in \mathbb{R}^{d_2}$ are two independent high-dimensional random vectors, each following a multivariate normal distribution:

$$\mathbf{y} \sim \mathcal{N}(\boldsymbol{\mu_1}, \boldsymbol{\Sigma_1}), \quad \mathbf{u} \sim \mathcal{N}(\boldsymbol{\mu_2}, \boldsymbol{\Sigma_2}). \tag{6}$$

While the true probability distributions of $\mathbf{y}$ and $\mathbf{u}$ are unknown due to their nature as natural language outputs and image watermarks, respectively, we assume them to follow multivariate normal distributions. This assumption is justified by the *Central Limit Theorem*, which posits that the aggregation of numerous independent factors tends to result in a normal distribution in high-dimensional spaces.

Consider the vectors $\mathbf{y} \in \mathbb{R}^{d_1}$ and $\mathbf{u} \in \mathbb{R}^{d_2}$, which are independently sampled $B$ times. Let us denote the sampled data as $\mathbf{y}_1, \mathbf{y}_2, \ldots, \mathbf{y}_B$ and $\mathbf{u}_1, \mathbf{u}_2, \ldots, \mathbf{u}_B$. The sample covariance matrix between $\mathbf{y}$ and $\mathbf{u}$ is:

$$\mathbf{Cov}(\mathbf{y}, \mathbf{u}) = \frac{1}{B-1} \sum_{i=1}^{B} (\mathbf{y}_i - \overline{\mathbf{y}})(\mathbf{u}_i - \overline{\mathbf{u}})^\top, \tag{7}$$

where $\overline{\mathbf{y}} = \frac{1}{B}\sum_{i=1}^{B} \mathbf{y}_i$ and $\overline{\mathbf{u}} = \frac{1}{B}\sum_{i=1}^{B} \mathbf{u}_i$ are the sample means of $\mathbf{y}$ and $\mathbf{u}$, respectively. Owing to the independence of $\mathbf{y}$ and $\mathbf{u}$ and the linearity of expectation, the expectation of the sample covariance matrix is:

$$
\begin{aligned}
\mathbb{E}[\mathbf{Cov}(\mathbf{y}, \mathbf{u})] &= \mathbb{E}\left[\frac{1}{B-1}\sum_{i=1}^{B}(\mathbf{y}_i - \overline{\mathbf{y}})(\mathbf{u}_i - \overline{\mathbf{u}})^\top\right] \\
&= \frac{1}{B-1}\sum_{i=1}^{B}\mathbb{E}\left[(\mathbf{y}_i - \overline{\mathbf{y}})(\mathbf{u}_i - \overline{\mathbf{u}})^\top\right] \\
&= \frac{1}{B-1}\sum_{i=1}^{B}\left(\mathbb{E}[\mathbf{y}_i\mathbf{u}_i^\top] - \mathbb{E}[\mathbf{y}_i]\mathbb{E}[\overline{\mathbf{u}}^\top] - \mathbb{E}[\overline{\mathbf{y}}]\mathbb{E}[\mathbf{u}_i^\top] + \mathbb{E}[\overline{\mathbf{y}}\overline{\mathbf{u}}^\top]\right) \\
&= \frac{1}{B-1}\sum_{i=1}^{B}\left(\boldsymbol{\mu_1}\boldsymbol{\mu_2}^\top - \boldsymbol{\mu_1}\boldsymbol{\mu_2}^\top - \boldsymbol{\mu_1}\boldsymbol{\mu_2}^\top + \boldsymbol{\mu_1}\boldsymbol{\mu_2}^\top\right) \\
&= \mathbf{0}.
\end{aligned}
\tag{8}
$$

Next, we analyze the variance of the sample covariance matrix.

$$
\begin{aligned}
\mathrm{Var}(\mathbf{Cov}(\mathbf{y}, \mathbf{u})_{ij}) &= \mathrm{Var}\left(\frac{1}{B-1}\sum_{k=1}^{B}(y_{ik} - \overline{y}_i)(u_{jk} - \overline{u}_j)\right) \\
&= \frac{1}{(B-1)^2}\left[\sum_{k=1}^{B}\mathrm{Var}(X_k) + 2\sum_{1 \le k < \ell \le B}\mathrm{Cov}(X_k, X_\ell)\right] \\
&= \frac{1}{(B-1)^2}\left(B\,\sigma_{y_i}^2\sigma_{u_j}^2 + B(B-1)\,\xi_{ij}\right) \\
&\le \frac{1}{(B-1)^2}\left(B\,\sigma_{y_i}^2\sigma_{u_j}^2 + (B-1)\,\sigma_{y_i}^2\sigma_{u_j}^2\right) \\
&= \frac{(2B-1)\,\sigma_{y_i}^2\sigma_{u_j}^2}{(B-1)^2} \\
&\le \frac{3\,\sigma_{y_i}^2\sigma_{u_j}^2}{B-1},
\end{aligned}
\tag{9}
$$

where $\xi_{ij} = \mathrm{Cov}\big((y_{i1} - \overline{y}_i)(u_{j1} - \overline{u}_j), (y_{i2} - \overline{y}_i)(u_{j2} - \overline{u}_j)\big)$ captures the dependence introduced by the shared sample means and satisfies $0 \le \xi_{ij} \le \sigma_{y_i}^2\sigma_{u_j}^2/B$.

To formalize the presence of spurious correlations, we apply *Chebyshev's inequality* to each entry of the covariance matrix:

$$\mathbb{P}\big(\big|\mathbf{Cov}(\mathbf{y}, \mathbf{u})_{ij}\big| \geq t\big) \leq \frac{3\,\sigma_{y_i}^2\,\sigma_{u_j}^2}{(B-1)t^2}. \tag{10}$$

By selecting an appropriate threshold $t$, for instance $t = k\sqrt{\mathrm{Var}\big(\mathbf{Cov}(\mathbf{y}, \mathbf{u})_{ij}\big)}$ for some constant $k > 0$, we ensure that there exists a non-negligible probability that the sample covariance $\mathbf{Cov}(\mathbf{y}, \mathbf{u})_{ij}$ exceeds $t$. This leads to the emergence of significant spurious correlations between the $i$-th dimension of $\mathbf{y}$ and the $j$-th dimension of $\mathbf{u}$.

Therefore, even though $\mathbf{y}$ and $\mathbf{u}$ are independent, the sample covariance matrix $\mathbf{Cov}(\mathbf{y}, \mathbf{u})$ can exhibit significant non-zero entries with a probability bounded away from zero when the number of samples $B$ is small. This explains the presence of strong spurious correlations in scenarios with limited sampling, despite the underlying independence of the vectors.

## B. Datasets

As described in the paper, we partition each dataset in a 6:4 ratio into a fine-tuning dataset ($\mathcal{D}_f$) and a probing dataset ($\mathcal{D}_p$). The probing dataset is further split into training, validation, and testing subsets in a 6:2:2 ratio. Table 9 presents the statistics of these datasets.

*Table 9.* Statistics of fine-tuning and probing datasets used in experiments.

| Dataset | Fine-Tuning ($\mathcal{D}_f$) | Probing ($\mathcal{D}_p$) | | |
| --- | --- | --- | --- | --- |
| | | Training | Validation | Testing |
| COCO | 70972 | 28389 | 9463 | 9463 |
| GQA | 89312 | 35725 | 11909 | 11908 |
| OCR-VQA | 124540 | 49817 | 16606 | 16605 |
| TextVQA | 15071 | 6029 | 2010 | 2009 |
| VisualGenome | 64948 | 25981 | 8660 | 8660 |

## C. Task-Irrelevant Private Content

We provide all the generated private content that we used in Table C. For the fine-tuning dataset, each sample in $\mathcal{U}_1$ is selected for embedding with equal probability. For the probing dataset, all samples in $\mathcal{U}_1$ and $\mathcal{U}_2$ are equally likely to be selected for embedding.

*Table 10.* All generated task-irrelevant private content, where *usernames* (bold) are embedded in both fine-tuning datasets and probing datasets, while *user_ids* are embedded only in fine-tuning sets.

| Subsets | Content |
| --- | --- |
| $\mathcal{U}_1$ | **username: Carlos Diaz**, user_id: 5374982160
**username: Sophia Chen**, user_id: 8250947613
**username: Ibrahim Al-Salem**, user_id: 9823046571
**username: Ava Murphy**, user_id: 4147285690
**username: Elena Mikhaylova**, user_id: 3759408621 |
| $\mathcal{U}_2$ | **username: Maximilian Schmidt**, user_id: 6473920581
**username: Vijay Sharma**, user_id: 9073264815
**username: Kim Jisoo**, user_id: 7568210945
**username: John Doe**, user_id: 1234567890
**username: Lucia Rodriguez**, user_id: 8397162045 |

*Table 11.* Average batch cosine gradient similarity comparison between original and modified samples on Qwen-VL for COCO with different privacy subset sizes.

| Origin | w/Privacy (Subset = 5) | w/Privacy (Subset = 100) | ImageTransf. | TextTransf. |
|---|---|---|---|---|
| 100.0 | 97.0 | 93.2 | 93.8 | 49.4 |

## D. Data Transformation

In Table 3, we compared the changes in training gradients of LLMs after incorporating privacy with the changes in gradients resulting from performing data transformation separately in the text and image modalities. In this section, we provide a detailed description of the data transformation.

### D.1. Image Transformation

We adopt a simple rule-based image transformation pipeline. For each image in the batch:

- We randomly rotate it by an angle in the range of $[-30°, +30°]$.
- With a probability of $50\%$, we perform a horizontal flip.
- We randomly adjust brightness in the range $[0.8, 1.2]$.
- We randomly adjust contrast in the range $[0.8, 1.2]$.

### D.2. Text Transformation

We employ GPT-4 to generate paraphrases of the existing question-answer pairs in our dataset. Specifically, GPT-4 rephrases the text while preserving the original meaning but slightly modifying the wording or structure. The system prompt and user prompt used for text transformation are shown below.

> **System Prompt:** You are a helpful assistant that carefully modifies text while preserving the original meaning. You will only replace or slightly alter one or two words with synonyms, ensuring minimal change. Do not alter the text structure or meaning beyond this. If the text starts with ⟨image⟩, keep that part exactly as is and do not remove or alter ⟨image⟩ in any way.
> **User Prompt:** Original text: {rest_part} Rewrite it by changing only one or two words to synonyms without any other words. Do not add any unrelated content.

### D.3. Examples

To offer a more intuitive illustration, we provide an example randomly selected from the COCO dataset in Figure 7. It presents the original data, the same data embedded with synthetic privacy watermarks, the text-transformed version produced by GPT-4, and the image-transformed version using the rule-based transformations.

## E. Impact of Privacy Subset Size

We perform an additional ablation study where we increase the number of items within each subset from 5 to 100. Specifically, we ask GPT-4 to generate 100 distinct usernames and corresponding user_ids for each subset, respectively. To avoid repetition, we request GPT-4 to check for duplicates after each generation. We use these 100 private items on Qwen-VL for COCO. Results are shown in Table 11. As the privacy subset size increases, the gradients of MLLMs exhibit more significant deviations from the original gradient updates, indicating that MLLMs spend more effort in each gradient step learning different privacy information when increasing the privacy subset size.

| Origin | w/ Privacy | Image Augmentation | Text Augmentation |
|---|---|---|---|
| 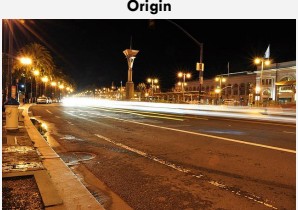 | 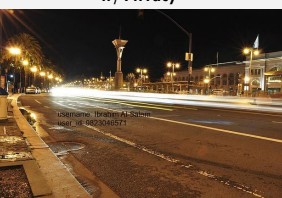 | 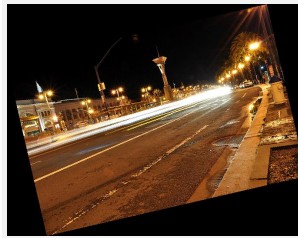 | 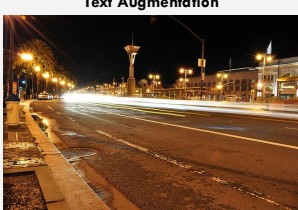 |

**Origin**

**Filename:** 000000341431.jpg

**Question:** What time of day does the image capture?
**Answer:** The image captures a night scene.

**Question:** How is the street illuminated in the image?
**Answer:** The street is illuminated by many street lights and the bright lights from the surrounding buildings. Additionally, there is a time lapse effect causing blurred lights on the road.

**Question:** Is there any notable architecture in the image?
**Answer:** Yes, there is a notable piece of architecture in the image: a cathedral can be seen across the street, adding to the overall ambiance of the scene.

**w/ Privacy**

**Filename:** 000000341431.jpg

**Question:** What time of day does the image capture?
**Answer:** The image captures a night scene.

**Question:** How is the street illuminated in the image?
**Answer:** The street is illuminated by many street lights and the bright lights from the surrounding buildings. Additionally, there is a time lapse effect causing blurred lights on the road.

**Question:** Is there any notable architecture in the image?
**Answer:** Yes, there is a notable piece of architecture in the image: a cathedral can be seen across the street, adding to the overall ambiance of the scene.

**Image Augmentation**

**Filename:** 000000341431.jpg

**Question:** What time of day does the image capture?
**Answer:** The image captures a night scene.

**Question:** How is the street illuminated in the image?
**Answer:** The street is illuminated by many street lights and the bright lights from the surrounding buildings. Additionally, there is a time lapse effect causing blurred lights on the road.

**Question:** Is there any notable architecture in the image?
**Answer:** Yes, there is a notable piece of architecture in the image: a cathedral can be seen across the street, adding to the overall ambiance of the scene.

**Text Augmentation**

**Filename:** 000000341431.jpg

**Question:** What moment of day does the image capture?
**Answer:** The image captures a dark scene.

**Question:** How is the road illuminated in the image?
**Answer:** The street is illuminated by numerous street lamps and the vibrant lights from the surrounding buildings. Additionally, there is a time lapse effect causing blurred lights on the road.

**Question:** Is there any distinctive architecture in the image?
**Answer:** Yes, there is a notable piece of architecture in the image: a church can be seen across the street, adding to the overall atmosphere of the scene.

*Figure 7.* Examples of data used for computing gradient similarity from COCO.

