# OpenReview forum: "Watch Out Your Album! On the Inadvertent Privacy Memorization in Multi-Modal Large Language Models"
_ICML.cc/2025/Conference — ICML 2025 poster_

### Official Review · Reviewer_4Jjq · 2025-03-01

**Overall Recommendation:** 4

**Summary:**

This paper investigates how MLLMs inadvertently memorize privacy that is irrelevant to the training objectives. The authors introduce a layer-wise probing framework to test whether task-irrelevant privacy is embedded in images during fine-tuning. They provide a formal mathematical proof to demonstrate that MLLMs encode such privacy due to spurious correlations within mini-batch training. Through extensive experiments, they find that even though task-irrelevant privacy does not directly affect downstream task performance, it still leads to distinct representational patterns within the parameters. Additionally, they show that this memorization effect is more pronounced when training with smaller batch sizes.


## =================update after rebuttal==============

Thanks for the authors' responses. My concerns have been well addressed.

**Claims And Evidence:**

In Table 3, the gradient similarity between models trained with privacy-embedded data and those trained on original data remains relatively high compared to transformations in the image and text modalities. This suggests that while task-irrelevant privacy does influence training dynamics, its effect may not be as strong as natural modality transformations. A deeper discussion on this observation would be beneficial.

**Essential References Not Discussed:**

The authors consider a novel privacy leakage scenario that differs from previous research. However, the introduction of related work on PII, which is most relevant to the authors’ study, is still not comprehensive enough.

**Ethical Review Concerns:**

None.

**Experimental Designs Or Analyses:**

Yes. I have checked all experimental designs and analyses in Section 3 and 4.


* The experiments are conducted exclusively on VQA tasks, which remains unclear whether similar behaviors occur in other multi-modal tasks. It would be helpful to evaluate additional multi-modal scenarios about the generality of the issue.

* The authors mainly study 7B-scale MLLMs with LoRA fine-tuning. Have you considered whether larger MLLMs exhibit more severe privacy memorization due to their increased capacity?

* Have you experimented with techniques like membership inference or gradient inversion to assess whether this privacy leakage is practically extractable?

**Methods And Evaluation Criteria:**

Yes. The proposed probing-based framework effectively quantifies the extent of memorization by assessing layer-wise representational differences in response to previously seen and unseen privacy.

**Other Comments Or Suggestions:**

No.

**Other Strengths And Weaknesses:**

Other Strengths:

* The authors provide a formal mathematical proof to explain that mini-batch training can induce spurious correlations between task-irrelevant privacy and downstream objectives, which demonstrates that such memorization is not just an empirical anomaly but an inherent issue in training dynamics.

Other Weaknesses: See the comments above.

**Questions For Authors:**

See the comments above.

**Relation To Broader Scientific Literature:**

The authors uniquely explore the inadvertent memorization of task-irrelevant privacy in MLLMs, which differs from prior attempts that mainly focus on task-relevant privacy risks, and expands the scope of privacy concerns beyond conventional attack scenarios.

**Theoretical Claims:**

Yes. I have checked Section 2 and Appendix A.

---

> ### Author Rebuttal · Authors · 2025-04-01
>
> We thank the reviewer for the thoughtful feedback. Below we address the main points raised in this review.
>
> * * *
> > **Claims And Evidence (W1)**: Privacy-embedded data has less impact on training than natural modality transformations.
>
> We clarify that embedding task-irrelevant privacy significantly impacts gradient updates, as evidenced by gradient similarities substantially lower than random noise compared to the Column Origin, indicating that MLLMs distinctly attend to the embedded privacy content.
>
> Although gradient similarity remains higher than with transformations in the text modality, it closely aligns with image modality transformations, especially in Qwen. This similarity is reasonable because task-irrelevant privacy embeddings introduce subtle perturbations intentionally designed to be unrelated to the downstream task, whereas natural transformations in images and texts directly affect task-relevant features, capturing greater attention from MLLMs.
>
> * * *
> > **Experimental Designs Or Analyses (W1)**: Findings may not generalize beyond VQA and broader multi-modal evaluation is needed.
>
> We choose VQA because it is one of the most widely adopted benchmark tasks for both training and evaluating MLLMs. Moreover, it involves both visual and textual reasoning in a relatively balanced manner, which serves as a strong representative for examining potential and inadvertent memorization.
>
> As the reviewer worries about the generalizability issue, we conduct additional experiments on LLaVA 1.5 7B using the image captioning task COCO_2014. In this setup, the gradient similarity between two runs on the original data is $97.8 \pm 5.4$, whereas the gradient similarity between the original data and the privacy-embedded data is $91.4 \pm 7.7$. **This aligns with our findings on VQA tasks and provides further evidence that MLLMs consistently exhibit the capability to encode task-irrelevant privacy across various downstream tasks.**
>
> * * *
> > **Experimental Designs Or Analyses (W2)**: Consider whether larger MLLMs exhibit more severe privacy memorization?
>
> We have investigated whether larger MLLMs exhibit more severe privacy memorization due to their increased capacity. We conduct additional experiments by (1) increasing the parameter scale from 7B to 13B, and (2) increasing the LoRA rank from 128 to 256. Due to space limitations, detailed results can be found in the response of *Experimental Designs Or Analyses (W1) to Reviewer 1hkx*.
>
> * * *
> > **Experimental Designs Or Analyses (W3)**: Consider testing extractability with membership inference or gradient inversion.
>
> **Yes, we have explored the possibility of privacy leakage via MIA, which shows minimal performance.** For space constrain, comprehensive analyses and results are provided in the response of *Relation To Broader Scientific Literature (W1) to Reviewer y5xM*.
>
> Regarding gradient inversion, it is particularly suitable for federated learning. **Our gradient similarity experiments revealed that MLLMs indeed capture weak signals during training. This suggests the potential feasibility of extracting task-irrelevant private content via gradient inversion within federated learning contexts.**
>
> * * *
> > **Essential References Not Discussed (W1)**: Extension of related work on PII.
>
> We have discussed several notable studies that investigate the potential to memorize PII. Specifically, we cited Carlini et al., who first revealed that LMs such as GPT-2 could emit PII in training samples. Lukas et al. further analyzed the leakage of PII in LLMs via black-box extraction.
>
> We further survey recent studies that explore PII memorization. Kim et al. introduced a probing tool that enables data subjects to detect the potential leakage of PII in LLM services. Shao et al. analyzed how the association capabilities of LLMs could facilitate privacy leakage. Meng et al. proposed a two-step attack to recover masked PII from training data. Recent research has also begun extending PII detection to MLLMs, such as evaluating autonomous web agents (Zharmagambetov et al.).
>
> **However, these prior works do not sufficiently consider scenarios where PII is entirely irrelevant to the training task.** We will incorporate these additional references in the related work section of the final version.
>
> [1] Carlini N, et al. Extracting training data from large language models. 2021.
>
> [2] Lukas N, et al. Analyzing leakage of personally identifiable information in language models. 2023.
>
> [3] Kim S, et al. Propile: Probing privacy leakage in large language models. 2023.
>
> [4] Shao H, et al. Quantifying Association Capabilities of Large Language Models and Its Implications on Privacy Leakage. 2024.
>
> [5] Meng W, et al. RR: Unveiling LLM Training Privacy through Recollection and Ranking. 2025.
>
> [6] Zharmagambetov, A, et al. AgentDAM: Privacy Leakage Evaluation for Autonomous Web Agents. 2025.
>
> * * *
> We hope that our explanations above can clarify your doubts and you can consider our work more favorably.

---

### Official Review · Reviewer_fXHf · 2025-03-10

**Overall Recommendation:** 3

**Summary:**

The paper explores the effects of incorporating synthetic task-irrelevant private content into training datasets on multimodal large language models (MLLMs). The authors analyze how such content influences gradient updates, model memorization, and the ability to differentiate between injected private information and standard task data. They conduct controlled experiments on multiple VQA datasets (COCO, GQA, OCR-VQA, etc.) and propose a probing method to verify whether task-irrelevant information is being inadvertently memorized by models. Key findings suggest that task-irrelevant private content can subtly alter model learning and performance, potentially leading to the unintended encoding of privacy-sensitive information.

**Claims And Evidence:**

The authors make several claims regarding the impact of task-irrelevant private content on model learning and memorization:

1. The presence of such content affects training gradients.
2. Models trained with private content exhibit a higher likelihood of responding to related test-time queries.
3. Intermediate embeddings in the trained model contain distinguishing information about the injected content.

These claims are supported by empirical results, but certain aspects lack strong validation. For instance, the influence of subset size on probing effectiveness is not thoroughly analyzed. Additionally, while Table 2 suggests significant performance degradation in OCR-VQA, TextVQA, and Visual Genome due to private content embedding, Section 4.2 claims a more generalized minimal effect, which appears contradictory. Addressing these inconsistencies would strengthen the claims.

**Essential References Not Discussed:**

N/A

**Experimental Designs Or Analyses:**

While the experimental setup is comprehensive, certain areas need further validation:

1. The size-dependent effect of private content injection on probing results remains unclear.
2. The observed performance decrease in Table 2 suggests varying levels of susceptibility across datasets. The authors should investigate why OCR-VQA and TextVQA experience higher accuracy drops.
3. The probing technique focuses on detecting indirect memorization but does not analyze whether private content can be specifically extracted with targeted prompts.

Addressing these issues would refine the paper’s experimental soundness.

**Methods And Evaluation Criteria:**

The methodology is well-structured, leveraging VQA datasets with a clear partitioning strategy for fine-tuning and probing.
However, the decision to use only five items of private content per subset may limit the probing method’s applicability.
A more detailed ablation study on the effect of subset size would enhance understanding.
The probing evaluation is novel, but it does not sufficiently address whether the model can explicitly regenerate private content under adversarial prompting, a key concern for privacy risks.

**Other Comments Or Suggestions:**

The authors should provide a deeper analysis of how subset size affects memorization sensitivity. Additionally, an experiment explicitly testing whether private content can be reconstructed via tailored adversarial prompts would improve the discussion on practical privacy risks.

**Other Strengths And Weaknesses:**

Strengths:

1. Introduces an interesting privacy-oriented perspective on task-irrelevant content in training data.
2. Provides novel probing methods to analyze memorization effects.
3. Experimental results are well-organized and provide useful insights into the impact of private content.

Weaknesses:

1. The probed memorization may not align with the real-world privacy risks community members are most concerned about (i.e., explicit memorization rather than influence on gradients).
2. The limited size of private content subsets in experiments may affect the generalizability of conclusions.

**Questions For Authors:**

1. Subset Design: Given that each subset for probing contains only five private content items, could this artificially amplify differentiation between the models? How does the number of private content items per subset influence the probing outcome?
2. Performance Drop: Table 2 suggests that OCR-VQA, TextVQA, and Visual Genome are disproportionately affected by private content injection. Could you clarify why these datasets exhibit greater sensitivity compared to COCO or GQA?
3. Explicit Memorization Risk: While the probing method demonstrates indirect memorization effects, have you tested whether models exposed to private content can regenerate it when prompted adversarially? Would such an evaluation align better with real-world privacy risks?

**Relation To Broader Scientific Literature:**

N/A
I'm not quite familiar with this area.

**Theoretical Claims:**

No.

---

> ### Author Rebuttal · Authors · 2025-04-01
>
> We thank the reviewer for the thoughtful feedback. We first summarize all the issues and suggestions raised by the reviewer, and address the main points raised in this review.
>
> * * *
> > **Issue 1**: While Table 2 suggests significant performance degradation in OCR-VQA, TextVQA, and Visual Genome due to private content embedding, Section 4.2 claims a more generalized minimal effect, which appears contradictory.
>
> **We argue that this arises primarily from the randomness inherent in the fine-tuning processes, rather than from privacy injection.** In most of our evaluations, the performance fluctuations induced by privacy embedding remain within a tolerable range—some results increase slightly, while others decrease, but no catastrophic deterioration occurs. This consistency suggests that the injected privacy does not substantially alter the core training data distribution or compromise its quality.
>
> Regarding the "greater sensitivity", we believe there are two main reasons:
>
> 1.**Fine-tuning data volume**: TextVQA and Visual Genome contain considerably fewer samples compared with COCO and GQA as shown in Table 5, thus their performance can exhibit larger variance.
>
> 2.**Randomness in the fine-tuning process**: The fine-tuning procedure for LLaVA appears to induce substantially more variability than that of Qwen-VL. Qwen-VL shows only minimal performance fluctuations across all five evaluated datasets when injected privacy. This implies that fine-tuning procedure, but not privacy injection can strongly affect the downstream performance.
>
> * * *
> > **Issue 2**: The decision to use only five items of private content per subset may limit the probing method's applicability. A more detailed ablation study on the effect of subset size would enhance understanding.
>
> In response, we have performed an additional ablation study where we increased the number of items within each subset from 5 to 100. Specifically, we ask GPT-4 to generate 100 distinct usernames and corresponding user_ids for each subset, respectively. To avoid repetition, we request GPT-4 to check for duplicates after each generation. We use these 100 private items on Qwen-VL for COCO. Results are shown below:
>
> |Origin|w/Privacy (Subset = 5)|w/Privacy (Subset = 100)|ImageTransf.|TextTransf.|
> |-:|-:|-:|-:|-:|
> |100.0|97.0|93.2|93.8|49.4|
>
> As the privacy subset size increases, the gradients of MLLMs exhibit more significant deviations from the original gradient updates, indicating that **MLLMs spend more effort in each gradient step learning different privacy information when increasing privacy subset size**.
>
> * * *
> > **Issue 3**: The probing evaluation is novel, but it does not sufficiently address whether the model can explicitly regenerate private content under adversarial prompting, a key concern for privacy risks.
>
> We have conducted comprehensive experiments to assess the effectiveness of general attack methods such as adversarial prompting and other practical methods like MIAs.
>
> We directly ask for the username visible in the testing image (Have you seen the username before?), and ask for the corresponding user_id that requires multi-hop reasoning (What is the user_id of the username in this image?). We find that both LLaVA and Qwen-VL perform nearly random accuracy (~50%) in the first scenario, and the accuracy for correctly identifying the user_id is 0%, which means that **direct prompting is ineffective in detecting the slight task-irrelevant privacy leakage**.
>
> Additionally, we have explored the possibility of privacy leakage via MIA. **Our evaluations reveal that MIAs such as LOSS, Zlib Entropy, and Min-k% Prob show only minimal improvements compared to MLLMs before fine-tuning.** For space constrain, comprehensive analyses and results are provided in the response of *Relation To Broader Scientific Literature (W1) to Reviewer y5xM*.
>
> * * *
> > **Issue 4**: The probed memorization may not align with the real-world privacy risks community members are most concerned about (i.e., explicit memorization rather than influence on gradients).
>
> Although explicit memorization scenarios such as direct prompting do not pose privacy risks in our setting, our work still identifies two significant real-world privacy risks that community members are concerned about.
>
> Firstly, similar to MIAs, an attacker with prior knowledge of a general range of usernames could randomly embed these privacy into images, and use our proposed probing method to detect which usernames were used in the fine-tuning, thus exposing sensitive information.
>
> Secondly, our findings demonstrate that MLLMs capture privacy-related information in gradients during mini-batch training, which provides theoretical support for gradient inversion attacks in federated learning settings. A malicious client could potentially exploit gradient information to infer task-irrelevant private content.
>
> * * *
> We hope that our explanations above can clarify your doubts and you can consider our work more favorably.

---

### Official Review · Reviewer_y5xM · 2025-03-11

**Overall Recommendation:** 4

**Summary:**

The paper examines how MLLMs inadvertently memorize task-irrelevant private content due to spurious correlations during mini-batch training. It begins with a preliminary analysis that formalizes the conditions under which such memorization occurs, followed by a rigorous mathematical proof demonstrating how task-irrelevant content can influence model parameters. To empirically validate their claims, the authors introduce a probing that embeds random privacy into images at varying rates and later tests whether the hidden states of each layer can distinguish between seen and unseen privacy after fine-tuning. They find that introducing randomly generated task-irrelevant privacy significantly shifts gradient directions compared to training without such private content. They also show that even though downstream task performance remains largely unaffected, MLLMs start encoding these spurious signals at lower layers. Finally, smaller batch sizes exacerbate this inadvertent memorization by increasing the likelihood of spurious correlations in mini-batches, which further strengthens the proof.

## update after rebuttal
I will keep my ratings since most of my concerns are solved.

**Claims And Evidence:**

The authors provide multiple lines of evidence supporting their claims about inadvertent memorization. First, the gradient similarity experiments demonstrate that introducing task-irrelevant private shifts training updates beyond random fluctuations, indicating it indeed encoding spurious signals. Second, the probing experiments show that MLLMs do distinguish between seen and unseen private content, suggesting that these signals are retained in internal representations despite having no direct impact on downstream tasks. Third, the ablation experiment on batch size demonstrates that spurious correlations in encoding arise from the training paradigm of mini-batch training. However, in the gradient similarity experiment, the authors report that the gradient similarity with privacy-embedded data remains notably high when compared to the baseline, which raises concerns about the validity of their claims.

**Essential References Not Discussed:**

None

**Experimental Designs Or Analyses:**

Yes. The experimental designs and analyses are comprehensive. However, for the gradient similarity experiments, since real-world training involves multiple batches rather than single-step updates, the authors should verify if gradient differences persist over multiple training steps. Moreover, in the probing experiments, the authors mention directly prompting models about memorized privacy, could the authors clarify the probing accuracy obtained through the direct prompting?

**Methods And Evaluation Criteria:**

Yes, the proposed method and evaluation criteria are well-justified, I can easily capture how the probing framework effectively measures inadvertent memorization.

**Other Comments Or Suggestions:**

None

**Other Strengths And Weaknesses:**

All strengths and weaknesses have been addressed above.

**Questions For Authors:**

None

**Relation To Broader Scientific Literature:**

This authors extend prior findings on privacy risks by examining a relatively unexplored scenario of task-irrelevant privacy memorization. However, they only show the possibility of such inadvertent memorization, without verifying whether the memorized privacy can be exposed or how it might be mitigated.

**Theoretical Claims:**

Yes.

---

> ### Author Rebuttal · Authors · 2025-04-01
>
> We thank the reviewer for the thoughtful feedback. Below we address the main points raised in this review.
>
> * * *
> > **Claims And Evidence (W1)**: High gradient similarity with privacy-embedded data raises concerns about the claims.
>
> We clarify that embedding task-irrelevant privacy significantly impacts gradient updates, as evidenced by gradient similarities substantially lower than random noise compared to the Column Origin, indicating that MLLMs distinctly attend to the embedded privacy content.
>
> Although gradient similarity remains higher than in text transformation, it closely aligns with image modality transformations. This similarity is reasonable because task-irrelevant privacy embeddings introduce subtle perturbations intentionally designed to be unrelated to the downstream task, whereas natural transformations in images and texts directly affect task-relevant features, capturing greater attention from MLLMs.
>
> * * *
> > **Experimental Designs Or Analyses / Supplementary Material (W1)**: Verify if gradient differences persist over multiple training steps, not just single-step updates.
>
> We conduct additional experiments using LLaVA 1.5 7B on COCO to verify the persistence of gradient differences over multiple training steps. We measured gradient similarity after multiple updates across 1, 10, and 100 mini-batches:
>
> |Batch Updates|Origin|w/Privacy|ImageTransf.|TextTransf.|
> |-|-|-|-|-|
> |1|98.3|92.9|85.3|5.3|
> |10|97.5|91.6|83.3|0.6|
> |100|91.9|84.6|74.6|0.2|
>
> All transformed scenarios gradually decrease with the number of mini-batch updates. Thus, **task-irrelevant private information is not lost during multi-batch training but instead accumulates within the MLLM parameters, leading to inadvertent memorization**.
>
> * * *
> > **Experimental Designs Or Analyses / Supplementary Material (W2)**: Clarify probing accuracy from direct prompting in experiments.
>
> **We find that direct prompting is completely ineffective for detecting memorized privacy in this task-irrelevant scenario.** Specifically, we directly ask for the username visible in the testing image (Have you seen the username before?), and ask for the corresponding user_id that requires multi-hop reasoning (What is the user_id of the username in this image?). Both LLaVA and Qwen-VL perform nearly random accuracy (~50%) in the first scenario, and the accuracy for correctly identifying the user_id is 0%, which means that direct prompting is ineffective in detecting the slight task-irrelevant privacy leakage.
>
> * * *
> > **Relation To Broader Scientific Literature (W1)**: The authors show the possibility of such inadvertent memorization, without verifying whether the memorized privacy can be exposed or how it might be mitigated.
>
> We have conducted experiments to verify whether the memorized privacy can be exposed through existing attack methods. For direct prompting, we have provided the results in response to **Experimental Designs Or Analyses / Supplementary Material (W2)**.
>
> Additionally, we have constructed a suitable dataset for MIA by leveraging GPT-4 to randomly generate 20 distinct samples embedding each piece of privacy information, which contains 100 member and 100 non-member instances.
>
> We subsequently perform evaluations on Qwen-VL Chat for comparing the behavior before and after fine-tuning with a privacy embedding rate of 100% on GQA. We consider three popular MIA methods: LOSS [1], Zlib Entropy [2], and Min-k% Prob [3]. The results are presented below. It indicates **only a marginal increase in MIA accuracy after fine-tuning, which means that MIAs generally fail when facing such weak, task-irrelevant signals in this paper**.
>
> |Model|LOSS|Zlib Entropy|Min-k% Prob|
> |-|-|-|-|
> |Before Tuning|0.507|0.638|0.535|
> |After Tuning|0.499|0.633|0.532|
>
> However, this does not inherently imply that task-irrelevant privacy information is secure. **Our gradient similarity experiments indicate noticeable differences in gradients when MLLMs encode privacy, suggesting a potential risk for gradient inversion attacks in scenarios such as federated learning.**
>
> Concerning mitigation strategies, we identify spurious correlations captured during mini-batch training as the critical factor to inadvertent memorization. Therefore, increasing batch sizes or employing gradient accumulation can significantly reduce the likelihood of encoding privacy. **Our ablation experiments clearly demonstrate that increasing batch sizes reduces gradient differences before and after encoding, supporting our hypothesis that larger batches effectively mitigate the probability of capturing such spurious correlations.**
>
> [1] Yeom S, et al. Privacy risk in machine learning. 2018.
>
> [2] Carlini N, et al. Extracting training data from large language models. 2021.
>
> [3] Shi W, et al. Detecting pretraining data from large language models. 2023.
>
> * * *
> We hope that our explanations above can clarify your doubts and you can consider our work more favorably.

---

> > ### Comment · Reviewer_y5xM · 2025-04-07
> >
> > Solved my concerns. I will keep my ratings.

---

### Official Review · Reviewer_1hkx · 2025-03-14

**Overall Recommendation:** 3

**Summary:**

This paper demonstrates that MLLMs can inadvertently memorize private content entirely unrelated to their training tasks. The authors provide a rigorous mathematical proof explaining how mini-batches introduce spurious correlations, leading MLLMs to store even random private data. Through a novel probing method, they reveal that MLLMs internally distinguish between private content they have encountered and content they have not.

**Claims And Evidence:**

Figure 6 shows similar accuracy trends for both the direct username query and the multi-hop user-id query. However, the dimensionality-reduced visualizations in Figures 4 and 5 appear distinctly different. Is there a more fundamental explanation for why these lower-dimensional representations exhibit such dissimilar patterns, while the probing accuracy suggests a similar level of memorization?

**Essential References Not Discussed:**

In the gradient similarity experiments, which part of the MLLM's parameters was tested? Is inadvertent privacy more likely to be encoded in the LLM parameters or in the vision tower?

**Experimental Designs Or Analyses:**

I have carefully checked the experimental designs and analyses.

For experimental designs, the use of the probing method provides new insights into how MLLMs encode spurious information within parameters. For experimental analyses, The gradient similarity experiments offer a straightforward way to illustrate that MLLMs focus on extra content.

However, empirical experiments are only conducted on LLaVA 7B and Qwen-VL 7B, which does not fully explore how varying parameter scales might influence inadvertent memorization.

**Methods And Evaluation Criteria:**

While the proposed probing framework is innovative, the probing classifiers themselves can introduce certain biases. Control tasks could further confirm whether the classifier captures genuine memorization patterns rather than noise [1].

[1] Hewitt J, Liang P. Designing and Interpreting Probes with Control Tasks[C]//Proceedings of the 2019 Conference on Empirical Methods in Natural Language Processing and the 9th International Joint Conference on Natural Language Processing (EMNLP-IJCNLP). 2019: 2733-2743.

**Other Comments Or Suggestions:**

See the questions above.

**Other Strengths And Weaknesses:**

I have already included the key questions in other sections. There are no other questions.

**Questions For Authors:**

See the questions above.

**Relation To Broader Scientific Literature:**

This paper studies whether MMLMs memorize randomly generated private content that does not help with the training tasks at all. While previous scientific literature mainly focused on private content already aligned with model objectives, this work shows that MLLMs can also encode irrelevant private content through spurious correlations in mini-batch training.

**Theoretical Claims:**

I have checked the correctness of proofs for theoretical claims. The analysis in Section 2 is particularly enlightening. By laying out the definition of task-irrelevant content and presenting rigorous proof in Appendix A, the authors offer a clear mathematical foundation for why random content can be memorized during mini-batch training. The ablation studies further validate the proof, making the argument more convincing.

---

> ### Author Rebuttal · Authors · 2025-04-01
>
> We thank the reviewer for the thoughtful feedback. Below we address the main points raised in this review.
>
> * * *
> > **Claims And Evidence (W1)**: Probing accuracy is similar, but why do visualizations differ so much?
>
> We thank the reviewer for this insightful observation, which we also find an intriguing phenomenon. The fundamental distinction between the two scenarios lies in how the MLLMs handle the queries:
> * In the direct username query scenario, the username is explicitly presented within the probing image, allowing the MLLM to directly recall the seen private content from memory.
> * In contrast, the user-id query scenario necessitates multi-hop reasoning, where the MLLM must infer the user-id by associating it with the username previously encountered during training.
>
> **Therefore, we argue that the second scenario inherently contains more nonlinear yet discriminative features. These complex nonlinear relationships are less distinctly captured by low-dimensional visualization, which results in less pronounced visual clusters.** However, the probing classifier's consistently high accuracy indicates that despite their subtlety in visualizations, these nonlinear relationships remain strongly distinguishable in the high-dimensional parameter space.
>
> * * *
> > **Methods And Evaluation Criteria (W1)**: Probing classifiers may be biased, control tasks can validate true memorization signals.
>
> We share the reviewer's concern regarding the potential biases introduced by probing classifiers. In this paper, we deliberately select the simplest linear classifier as the probing model to minimize the bias, following the method proposed by Hewitt et al.
>
> Moreover, we construct control tasks by randomly shuffling labels in accordance with Hewitt et al. We find that the test accuracy consistently maintains around 50%, aligning with random guessing. This confirms that subtracting the control task accuracy (i.e., using selectivity) and simply measuring the probing accuracy lead to similar conclusions.
>
> [1] Hewitt J et al., Designing and Interpreting Probes with Control Tasks, 2019.
>
> * * *
> > **Experimental Designs Or Analyses (W1)**: Experiments on 7B models only, scale effects on memorization remain unexplored.
>
> We thank the reviewer for raising this concern regarding the impact of parameter scales. In response, we conduct additional experiments by (1) increasing the parameter scale of LLaVA from 7B to 13B, and (2) increasing the LoRA rank from 128 to 256 in the LLaVA 7B setting.
>
> Our findings indicate that **when privacy is embedded of different parameter scales, the gradients obtained from privacy maintain significant divergence from those of normal training**. Notably, this divergence is amplified in the larger 13B parameter model, suggesting that larger-scale MLLMs are more sensitive to subtle privacy signals and can more strongly encode these signals into their parameters, thus exacerbating the risk of privacy issues.
>
> * Results for LLaVA 13B
>
> |Dataset|Origin|w/Privacy|ImageTransf.|TextTransf.|
> |-|-:|-:|-:|-:|
> |coco|97.4|91.4|85.8|1.9|
> |gqa|91.8|81.5|74.2|1.2|
> |ocrvqa|98.0|73.8|28.8|1.3|
> |textvqa|96.7|90.6|67.1|2.4|
> |vg|89.1|78.8|73.5|2.9|
>
> * Results for LoRA-256 on LLaVA 7B
>
> |Dataset|Origin|w/Privacy|ImageTransf.|TextTransf.|
> |-|-:|-:|-:|-:|
> |coco|99.4|93.9|87.3|2.8|
> |gqa|98.2|86.8|76.9|1.8|
> |ocrvqa|98.8|77.0|30.4|2.8|
> |textvqa|99.4|94.6|72.4|2.0|
> |vg|97.6|87.0|75.6|2.6|
>
> * * *
> > **Essential References Not Discussed (W1)**: Unclear which parameters were tested, LLM or vision tower may differ in privacy encoding.
>
> In this paper, we initially follow the default settings of LLaVA 1.5 7B and Qwen-VL Chat, where for Qwen we freeze the entire vision block and apply LoRA only to the language transformer block, while for LLaVA we fine-tune both the vision and language blocks.
>
> To further investigate whether inadvertent privacy is more likely to be encoded in the LLM parameters or in the vision tower, we freeze all LLM parameters and allow the final layer of vision block to update its gradients during fine-tuning on COCO in Qwen-VL Chat. The results are presented below:
>
> |Dataset|Origin|w/Privacy|ImageTransf.|TextTransf.|
> |-|-:|-:|-:|-:|
> |Language Block|100.0|97.0|93.8|49.4|
> |Vision Block|100.0|32.6|20.5|51.8|
>
> Surprisingly, we observe a significant reduction in gradient similarity for the vision block when privacy is embedded into the images, while the gradient similarity in the text modality transformation remains relatively unchanged.
>
> **Thus, inadvertent privacy is more likely to be encoded in the vision tower**. We will include these experimental results in the final version to emphasize this heightened privacy concern.
>
> * * *
> We hope that our explanations above can clarify your doubts and you can consider our work more favorably.

---

### Decision · Program_Chairs · 2025-05-01

**Decision:**

Accept (poster)

**Comment:**

The paper studies unintentional memorization of task-irrelevant features in multimodal LLMs.
The paper shows that certain watermark signals that are uncorrelated with task labels can nevertheless be memorized by the model, and propose a theoretical explanation based on mini-batch training.

Reviewers generally agreed that this paper provides a valuable contribution to the literature on memorization in LLMs. I recommend acceptance.